# The Efficiency of Centrifugation in the Detection of Mollicutes in Bovine Milk

Anelise Salina [1] , Bruna Churocof Lopes [2], Fabiana Luccas Barone [1] and Helio Langoni [1,*]

1   Department of Animal Production and Preventive Veterinary Medicine, School of Veterinary Medicine and Animal Science, São Paulo State University, Botucatu 18618-681, Brazil; anelise.salina@unesp.br (A.S.); fabiana.barone@unesp.br (F.L.B.)
2   Independent Researcher, Botucatu 18618-681, Brazil; brunachurocoflopes@gmail.com
*   Correspondence: helio.langoni@unesp.br

**Abstract:** Mastitis is one of the main diseases that affects dairy cattle. It is characterized by a multifactorial disease with multiple etiologies including microorganisms such as bacteria, viruses, fungi, yeasts, and algae. Mastitis caused by *Mycoplasma* spp. results in a contagious nature of infection and has been studied much in Brazil. The objectives of this research were (1) to investigate in clinical mastitis milk samples, by conventional PCR, the presence of the following mycoplasma species: *Mycoplasma* (*M.*) *bovis*, *M. bovirhinis*, *M. bovigenitalium*, *M. californicum*, and *M. alkalescens* and (2) compare the PCR Mollicutes detection rates using previous concentration and homogenization steps of the milk samples. Of the 993 clinical mastitis milk samples analyzed, 13.7% (136/993) and 8.0% (80/993) were positive for the Mollicutes class after concentration and homogenization, respectively, and all of them were PCR-negative for the five species investigated. Of the 70 expansion milk tanks analyzed, all of them were PCR-negative for the Mollicutes class. The importance of the previous concentration of milk in the routine of molecular analysis was evidenced when compared to the results obtained only with homogenization.

**Keywords:** clinical mastitis; PCR; mammary mycoplasmosis; *Mycoplasma* spp.





## 1. Introduction

Mastitis is a disease of great concern in dairy cattle because its presence in herds results in economic losses and a loss of milk quality, being present in up to 23 to 40% of lactating cows [1,2]. Defined as inflammation of the mammary gland, it is caused by alterations in the glandular tissue, in addition to promoting cellular, physical–chemical, and organoleptic alterations in the milk [3]. The disease may have an infectious, physiological, traumatic, or allergic origin and may be of an infectious origin caused by more than 140 species of microorganisms, including contagious and environmental [4–6].

The contagious disease is characterized by animal-to-animal transmission during and after milking, and its most frequent pathogens are *Staphylococcus aureus*, coagulase-negative staphylococcus species, *Streptococcus agalactiae*, *Corynebacterium bovis*, and *Mycoplasma* spp. Its preferred habitat is the interior of the mammary gland and the skin surface of the teats. In the environmental ones, pathogens present in the dairy farm's environment are involved, such as *Escherichia coli*, *Klebsiella pneumoniae*, *Streptococcus uberis*, *Enterobacter* spp., and *Pseudomonas aeruginosa*, among others [7,8].

More than 25 species belonging to the class Mollicutes cause mastitis and other clinical manifestations in heifers and cows. They are more frequent in dairy herds: *M. californicum*, *M. alkalescens*, *M. arginini*, *M. bovigenitalium*, *M. canadense*, *M. dispar*, *M. bovirhinis* [9], and *M. bovis* [10]. Mastitis caused by *M. bovis* is the most prevalent among mycoplasma species, reported more frequently in herds with more than 500 lactating animals [11,12]. The agent is also responsible for other diseases in animals, such as pneumonia and arthritis in calves and heifers, as well as reproductive and fertility problems in adults [13]. Although *M. bovis*

has a higher occurrence in mycoplasma mastitis around the world, other species may occur concomitantly with it [14,15]. Clinical signs in cows such as arthritis and pneumonia may suggest the presence of *M. bovis* in the herd [12]. In Brazil, the presence of the pathogen has been detected in dairy herds, in cases of clinical mastitis, and in bulk tanks [16–18].

Mastitis by mycoplasma is highly contagious and usually affects more than one breast quarter with a significant drop in production. *Mycoplasma* species are refractory to conventional treatments and are characterized by purulent mastitis, lack of odor, and discolored secretions, and in some cases, cows may not have any evident clinical changes, even in severe cases [12]. Most of the clinical cases become chronic with loss of function of the teats involved [19].

The identification of the pathogen can be performed by microbiological culture, which is commonly used; however, it has some limitations, such as the long incubation period [20]. In cases of co-infection with other mycoplasma species, isolation and identification become difficult using this method, making it necessary to use alternatives such as molecular techniques to identify pathogens [10,21]. What makes its differentiation by microbiological culture even more complex is the occurrence of co-infection by more than one species of mycoplasma, and studies estimate that in about 50% of positive cows, the infection is caused by more than one species [15].

Thus, given the occurrence of mammary mycoplasmosis in herds around the world and specifically Brazil as well as the importance of its detection for the implementation of control programs, the aim of the present study was (1) to investigate five species of mycoplasma in clinical bovine mastitis and (2) to compare the Mollicutes PCR results in the previous concentration and homogenization steps of the milk samples.

## 2. Materials and Methods

This work was approved by the Ethics Committee of the School of Veterinary Medicine and Animal Science (CEUA) Botucatu City, São Paulo state, Brazil under protocol 0136/2017, as part of a thematic research project supported by FAPESP (2015/19688-8).

The Tamis test was used to detect clinical mastitis cases, and milk collection was performed in animals that had visible changes in the milk, as well as systemic clinical signs of inflammation in the breasts such as edema, heat, pain, and redness. A total of 993 samples of around 15 mL of milk from mammary quarters with clinical mastitis signs were collected aseptically and kept frozen at $-20\,^{\circ}$C on the properties until they were sent to the laboratory for analysis. Seventy milk samples from bulk tanks were collected every two weeks from ten different dairy farms in Brazil, located in São Paulo, Minas Gerais, and Paraná states.

All these herds have a mastitis control program, somatic cell count records (SCC < 400,000 cells/mL) [22], high-production Holstein cows (>20 L/cow/day), and at least 200 lactating cows under mechanical milking.

All clinical milk samples obtained for this study were convenience samples, previously analyzed for the research of other microorganisms under the objectives of the thematic project supported by FAPESP. All samples analyzed in this study showed no growth on conventional microbiological culture, plated on 5% blood agar and Mac Conkey agar.

### 2.1. DNA Extraction

The preparation of the milk samples took place in two stages, one stage of homogenization and the other of a previous centrifugation of each of the samples.

Aliquots of approximately 15 mL of milk were homogenized manually with a pipette, and 1 mL was placed in a new microtube for carrying out DNA extraction. The remaining volume, approximately 14 mL, was centrifuged at $5000\times g$ for 30 min to concentrate the agent in the milk sample, adapted according to Punyapornwithaya et al. [23] with a minor modification as follows: after centrifugation, the supernatant was removed using a pipette and a swab, and the obtained pellet was resuspended in 1 mL of buffered saline solution, pH 7.2. Subsequently, both 1 mL milk aliquots were individually submitted to

DNA extraction using the thermolysis technique, according to Fan et al. [24], with the following adjustments: initial centrifugation was performed at 12,000× *g* for 10 min, then the supernatant containing fat and proteins was discarded, keeping the pellet at the bottom of the microtube. Then, 700 µL of buffered saline solution, pH 7.2, was added to the pellet, with new centrifugation at 12,000× *g* for 10 min. Once again, the supernatant was discarded and 50 µL of ultrapure water was added to the pellet. Then, the samples were kept in a water bath at 100 °C for 10 min and then immediately submerged in ice for five minutes. As a final step, we performed centrifugation at 12,000× *g* for 10 min and recovery of the supernatant with the DNA in elution.

### 2.2. Conventional PCR

For the initial screening of milk samples, the universal primer pair for the Mollicutes class was used according to Van Kuppeveld et al. [25]: GPO-3 (5′-GGGAGCAAACAGGATT-AGATACCCT-3′) and MGSO (5′-TGCACCATCTGTCACTCTGTTAACCT-3′) aimed to identify a product of 270 base pairs (bp).

After they were determined as PCR-positive with generic primers for the Mollicutes class, DNA amplification of the species of *M. bovis*, *M. alkalescens*, *M. bovigenitalium*, *M. bovirhinis*, and *M. californicum* was carried out individually with specific primers, using the sequences described in Table 1 above. All the amplifications were performed in a model Mastercycler Gradient (Eppendorf®).

**Table 1.** Mycoplasma species, oligonucleotide sequences, amplification cycles, and references used to perform conventional PCR in bovine clinical mastitis milk samples. Botucatu, São Paulo, Brazil, 2023.

| Mycoplasma Species | Oligonucleotides Sequence | PCR Conditions | Amplicon Size (Base Pair) | Reference |
|---|---|---|---|---|
| *Mycoplasma bovis* | MboF (5′-CCTTTTAGAT-TGGGAT-AGCGGATG-3′) and MboR (5′-CCGTCAAGGTAGCAT-CATTTCCTAT-3′) | Pre-heating at 94 °C/3 m; 35 cycles in denaturation at 94 °C/1 m, annealing at 60 °C/1 m, extension at 72 °C/1 m; final extension 72 °C/30 m. | 360 bp | Chávez González et al. [26] |
| *Mycoplasma alkalescens* | MakF (5′-GCTGTTATAGGGA-AAGAAAACT-3′) and MakR (5′-AGAGTCCTCGA-CATGACTCG-3′) | Pre-heating at 94 °C/9 m; 40 cycles in denaturation at 94 °C/30 m, annealing at 60 °C/1 m, extension at 72 °C/1 m; final extension at 72 °C/7 m. | 704 bp | Kobayashi et al. [27] |
| *Mycoplasma bovigenitalium* | MbgF (5′-CGTAGAT-GCCGCATGGCATTT-ACGG-3′) and MbgR (5′-CATTCAATATAG-TGGCATTTCCTAC-3′) | Pre-heating at 94 °C/9 m; 35 cycles in denaturation at 94 °C/30 m, annealing at 60 °C/1 m, extension at 72 °C/1 m; final extension 72 °C/7 m. | 312 bp | Kobayashi et al. [27] |
| *Mycoplasma bovirhinis* | MbrF (5′-GCTGA-TAGAGAGGTCTATCG-3′) and MbrR (5′-ATTACT-CGGGCAGTCTCC-3′) | Pre-heating at 94 °C/9 m; 35 cycles in denaturation at 94 °C/30 m, annealing at 60 °C/1 m, extension at 72 °C/1 m; final extension 72 °C/7 m. | 316 bp | Kobayashi et al. [27] |
| *Mycoplasma californicum* | McF (5′-GCACTTAGAC-GAAAGAGGGATT-3′) and McR (5′-GATTATC-ATCACCTTTGGGACT-3′) | Pre-heating at 95 °C/15 m; 45 cycles in denaturation at 94 °C/15 s, annealing at 60 °C/1 m, extension at 72 °C/30 s; final extension at 72 °C/5 m. | 280 bp | Boonyayatra et al. [28] |

The positives controls used in this study were *M. bovis* ATCC 25553, *M. alkalescens* ATCC 29103, *M. bovigenitalium* ATCC 19852, *M. bovirhinis* ATCC 27748, and *M. californicum* ATCC 33461. The resulting PCR products were subjected to horizontal electrophoresis on a 1.5% agarose gel in boric acid-Tris-EDTA buffer and developed with Nancy-520. The

DNA fragments were analyzed comparatively with DNA markers of 100 base pairs, being analyzed and photographed in an image analyzer.

### 3. Results

Of the 993 samples of clinical mastitis cases, 13.7% (136/993) and 8.0% (80/993) were PCR-positive with the generic primers for the Mollicutes class, using the methods of concentration and homogenization of the milk, respectively. The previous concentration of milk samples had a 58.8% increased possibility to detect Mollicutes by conventional PCR.

The PCR results of the species *M. bovis*, *M. alkalescens*, *M. bovigenitalium*, *M. bovirhinis*, and *M. californicum* was negative in all clinical mastitis milk samples that were previously positive for the Mollicutes class.

Of the total of 70 milk samples from bulk tanks previously concentrated for DNA extraction, all were negative for the Mollicutes class and, therefore, were not tested for any of the species proposed in this study.

### 4. Discussion

According to González et al. [29], 30 to 40% of samples from bulk tanks with positive cases for mycoplasmas on the property are negative for *M. bovis* and may be erroneously considered free of the microorganism. This fact occurs because the milk of infected cows is diluted in a large volume, and the pool of milk from positive teats for the agent alongside milk from healthy cows results in a concentration below the minimum level of detection of the microorganism, according to Biddle et al. [30]. The fact that milk from cows with clinical mastitis is discarded and not included in the bulk tank suggests that the results obtained in this study may be underestimated, leading to a misinterpretation of agent-free properties.

The Mollicutes class has a small size compared to other classes of bacteria, and the absence of a cell wall could make it difficult to decant cells into the pellet. In this way, we used the methodology described by Punyapornwithaya et al. [23], with the aim of increasing the detection of Mollicutes in conventional PCR. However, one limitation of the present study is the fact that it was not possible to evaluate higher speeds, which could not be achieved by the centrifugal equipment used.

In a study carried out by Higuchi et al. [31], it was demonstrated that the detection of *M. bovis* by the microbiological culture of the bulk tank milk ranged from 1.8% (3/166) to 6.1% (11/180), while the detection of *M. californicum* ranged from 6.8% (18/258) to 8.3% (3/50). Although the study presented here did not assess the viability of mycoplasma from the bulk tank using microbiological culture, the importance of the technique is proven when there is a need to detect these pathogens in bulk tank milk that is associated with PCR.

One of the obstacles in the diagnosis of mycoplasmas in a bulk tank is the dilution of the pathogen in whole milk, with a consequent decrease in the possibility of detection, which can result in a false-negative result in an analysis of a single milk sample. The suggestion to analyze samples from bulk tanks from the same property at intervals of fifteen days is based on a study carried out by Francoz et al. [32], who analyzed three samples from each herd studied in the period of one month and obtained a single PCR-positive bulk tank sample among the three sampled. According to Nicholas et al. [12], the elimination of mycoplasmas in mastitic milk can occur in up to $1 \times 10^6$ mycoplasmas/mL of milk, which could allow the detection of an infected animal among at least 1000 animals, but the authors point out that the greater the number of cows that make up the milk sample, the lower the detection sensitivity.

The positivity of Mollicutes in Brazilian herds can vary and was demonstrated in previous studies with 3.0% and 8.6% of clinical mastitis samples [16,17] and 16.4% of bulk tanks positive for this class of microorganisms [33]. In other countries, positivity is also variable: in the USA, it varies from 1 to 8% [34]; in Greece, to 5.4% [35]; in Thailand, to 1.8% [36]; and in Egypt, to 7.53% [37].

The scarcity of studies on mycoplasmosis in Brazil makes it difficult to discuss results obtained previously, restricting agreement with studies published in other countries. The

scarcity of studies on the occurrence of mycoplasma species in bovine mastitis in Brazil made us opt for methodologies and research of mycoplasma species found by authors in other countries. So far, our research group has already obtained results similar to previous results on bovine mastitis caused by *M. bovis* with the methodology applied in this present study [16,17]. The present study did not obtain positive results for the detection of *M. bovis* in clinical mastitis milk, which is considered one of the most prevalent species in herds from other countries, where, in the United States, for example, its prevalence can reach 75.1% of isolates of clinical mastitis [14].

The diagnosis of mycoplasma mastitis is usually made by microbiological culture; however, PCR is an alternative to promote a faster result with greater sensitivity [38]. The results obtained in this present study from previously centrifuged samples demonstrated the importance of this step for molecular diagnosis, with an increase in detection of the Mollicutes class from 8.0% to 13.7%. Although we did not carry out the initial quantification of mycoplasmas in the milk samples, it is suggested as an alternative to concentrating the milk sample, employing centrifugation and resuspension of the resulting pellet in saline solution. A similar result was observed by Punyapornwithaya et al. [23] which obtained an increase of up to four times in the identification of a positive sample, which may explain the greater positivity of this study obtained with the concentration of the agent in the milk samples.

As for all the negative results obtained in the investigation of mycoplasma species, this could be explained by the occurrence of *Ureaplasma* spp. or *Acholeplasma* spp.; however, they were not investigated. These microorganisms are considered opportunists in cases of mastitis and bulk tank milk and belong to the Mollicutes class, and they could be amplified with the primer used in the present study. One of the alternatives for differentiating between species of *Mycoplasma* and *Acholeplasma* is the use of digitonin to inhibit the growth of *Acholeplama* spp. in a culture medium [28] and analysis of the melting curve for genus confirmation [39].

**5. Conclusions**

The presence of the Mollicutes class was confirmed by PCR in cases of bovine clinical mastitis, but not in bulk tank milk.

The concentration technique increased the detection of the Mollicutes class when compared with only the homogenization.

**Author Contributions:** Conceptualization, A.S. and B.C.L.; methodology, A.S. and B.C.L.; formal analysis, A.S.; investigation, B.C.L.; resources, H.L.; writing—original draft preparation, A.S. and B.C.L.; writing—review and editing, A.S., F.L.B. and H.L.; funding acquisition, H.L. All authors have read and agreed to the published version of the manuscript.

**Funding:** This research was funded by FAPESP, grant number 2018/09086-9.

**Institutional Review Board Statement:** The animal study protocol was approved by the Ethics Committee of CEUA (protocol code 0136/2017 approved on 8 June 2017).

**Informed Consent Statement:** Not applicable.

**Data Availability Statement:** Not applicable.

**Acknowledgments:** The authors are grateful to the dairy farms for providing milk samples for this study.

**Conflicts of Interest:** The authors declare no conflict of interest.

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
