# Peer review of "The Efficiency of Centrifugation in the Detection of Mollicutes in Bovine Milk"

_ruminants, doi:10.3390/ruminants3030021_

Round 1

Reviewer 1 Report

Authors have addressed an important question and have described a molecular diagnostic approach. Detection of the infectious agent by PCR would be an important tool for early detection.  Detection by PCR have been hailed in many research groups however it remains limited by species identification.

  The manuscript carries the scientific merits however at several instances rephrasing of sentence would improve the understanding for the readers:

Line 12 - Of contagious …… I t may be rephrased as : “mastitis caused by Mycoplasma spp result in contagious nature of infection and have been studied much in Brazil”

Line 26 - Mastitis has a great importance in dairy. – This line should be corrected since the disease causes loss it cannot be important.

Authors must rephrase sentences at several places for clear understanding improve in the manuscript

2)  In the material and method section the authors must include the PCR reaction steps. Authors should mention the genomic regions selected for designing the primers along with the melting temperatures and length of the products.

3)PCR results if presented in table format besides authors must elaborate if “previously concentrated samples” affect the quality of samples as compared to fresh samples. It would be an important observation.

4) Authors mentioned M. bovirhinis and M. californicum in the abstract but have missed to mention their detection in result section

5) Discussion section is appropriate and it would have been better if the it included the discussion about the selection of molecular approach for the detection

 Overall, the manuscript has the scientific merit and it discussed about the molecular approach of detection of Mastitis. I recommendation this for publication.

Overall,manuscript needs improvement in quality of English. Some sentences has to be rephrased for better understanding.

Author Response

Authors have addressed an important question and have described a molecular diagnostic approach. Detection of the infectious agent by PCR would be an important tool for early detection.  Detection by PCR have been hailed in many research groups however it remains limited by species identification.

The manuscript carries the scientific merits however at several instances rephrasing of sentence would improve the understanding for the readers:

Line 12 - Of contagious …… I t may be rephrased as : “mastitis caused by Mycoplasma spp result in contagious nature of infection and have been studied much in Brazil”

Thanks to the reviewer for the suggestion. We rephrased and added the suggestion.

Line 26 - Mastitis has a great importance in dairy. – This line should be corrected since the disease causes loss it cannot be important.

Thanks for the suggestion, we replaced the previous sentence with: “Mastitis is a great concern disease in dairy cattle because their presence in herds results in economic losses and loss of milk quality, being present in up to 23 to 40% of lactating cows” (lines 26-28).

Authors must rephrase sentences at several places for clear understanding improve in the manuscript.

We did a general review of the manuscript and changed some points to improve the readers' understanding.

2)  In the material and method section the authors must include the PCR reaction steps. Authors should mention the genomic regions selected for designing the primers along with the melting temperatures and length of the products.

Thanks for the suggestion. For further clarification of the methodology used in our study, we have added a Table to the methodology (line 121).

3)PCR results if presented in table format besides authors must elaborate if “previously concentrated samples” affect the quality of samples as compared to fresh samples. It would be an important observation.

Thanks for the suggestion, we agree that the results presented in the tables are of better understanding for the readers and provide more information, when available. However, we chose to keep the format of the results as described in the manuscript because we do not have information about the quality of the samples when the suggested comparison is made.

4) Authors mentioned M. bovirhinis and M. californicum in the abstract but have missed to mention their detection in result section.

Thanks to the reviewer for the observation. In our manuscript, we did not obtain amplification of any of the five species we proposed. The sentence in the “results” section that explains this finding is described in lines 138-140: “The research of the species: M. bovis, M. alkalescens, M. bovigenitalium, M. bovirhinis and M. californicum, was negative in all clinical mastitis milk samples previously positive to the Mollicutes class”.

5) Discussion section is appropriate and it would have been better if the it included the discussion about the selection of molecular approach for the detection.

Thanks to the reviewer for the comment. We have added information about the choice of methodology in lines 176-181 of this manuscript. The choice of using prior centrifugation at 5000g was in accordance with the study carried out by (PUNYAPORNWITHAYA et al., 2009) while the selection of species chosen for our study was based on findings by other authors. The fact that in Brazil there are no studies on the research of species of Mycoplasma spp. occurring in bovine mastitis, except M. bovis, led us to opt for findings by authors from other countries.

Overall, the manuscript has the scientific merit and it discussed about the molecular approach of detection of Mastitis. I recommendation this for publication.

We thank the reviewer for his relevant suggestions and compliments and hope that we have answered your requests in the best way.

Reference

PUNYAPORNWITHAYA, V. et al. The effect of centrifugation and resuspension on the recovery of Mycoplasma species from milk. Journal of Dairy Science, v. 92, n. 9, p. 4444–4447, 2009.

Reviewer 2 Report

This manuscript presented herein by Salina and colleagues represents an interesting investigation on whether centrifugation of milk samples to concentrate potential contaminating microbes prior to DNA extraction can improve the detection of Mastitis-associated Mollicute Species in cow milk. While this is certainly an important topic and question to be addressed, I unfortunately found the methods used to be a bit lacking, and insufficient to appropriately address this question (see comments below). As such, the results may have been skewed, not necessarily due to the absence of these species, but rater improper handling of the specimen that could result in false negatives. It is unclear if these issues are simply with insufficient reporting of methods used, or if no alternative methods were used altogether. Significant improvements need to be made to the methods before this manuscript can enter a publishable state. Below are my comments and recommendations:

Major points:

1) The authors aim to assess the efficiency of centrifugation in improving detection of Mollicutes in bovine milk, but do not offer a comparison between different centrifugation set ups, and instead use only a single centrifugation speed and time (5000xg for 30 minutes line 97 of manuscript).

Many issues here:

1a) How was the speed and time of centrifugation decided? If the authors performed a comparison to determine optimal time and speed of centrifugation those data should appear in this manuscript to justify why these conditions were chosen.

1b) Mollicutes are particularly small, and not as dense as eubacteria given their lack of a cell wall. It is well understood within the Mycoplasmology community that speeds of roughly 10,000xg for at least 15 minutes at 4C are typically required to obtain sizeable pellets from centrifugation of many Mycoplasma species, and that is on liquid broth that is not as dense, complex, or as viscous as milk. It is likely, therefore, that if this was the only condition tested, that the centrifugation was insufficient to pellet and concentrate the mollicute species tested herein, potentially resulting in false negative results. If different centrifugation set ups were not assessed, this should be performed for the revision of this manuscript.

1c)The authors state in the methods (line 79), that milk samples were kept frozen at -20C prior to analysis. Mollicutes are particularly susceptible to loss of titer when frozen at temperatures below -80C (I have empirically tested this in the laboratory with multiple mycoplasma species and have found that when samples are frozen at -20C then thawed and assessed for viable counts you experience a reduction of roughly 8 logs of viable Mycoplasma counts vs a 1.5 to 2 log reduction when samples are frozen at -80C. If the mollicute cells were lysed due to storage conditions this could have further resulted in false negative detection results, especially since DNA definitely does not pellet with centrifugation at 5000xg for 30 minutes.

2) The other major issue I have is with the method of detection itself (i.e. PCR set up used in the study). The authors do not describe multiple key details about the method, for example:

2a) Is the PCR with the primers provided supposed to result in amplicons with size differences to determine sample positivity/negativity, or is it a matter of amplification vs no amplification?  What does a positive vs a negative result look like? Gel images should be provided as data along with whether the sample was deemed positive/negative to support the manuscript.

2b) What regions in the mollicute genomes do the primers amplify, and what are the expected amplicon sizes? What are the optimal PCR conditions used in these reactions? 

2c) How are the primers validated? A quick NCBI blast of the universal mollicute primers finds 100% covarage on multiple other families of bacteria including Escherichia species, Achromobacter, Bordetella, Acinetobacter, Vibrio, and Bacillus species, and these with E values of 0.004 or less, and 100% sequence coverage. In silico PCR produces multiple viable amplicons from non Mollicute species when the listed primers are used, with variable sizes of amplicons. Is this the same for the species specific primers? Some detailed information about the PCR conditions used, and what defines a positive vs a negative is critical here. Should these primers not be specific, it is possible the positive results obtained may be false positives. The authors should expand the methods to include how these primers are validated. Again, if this is a matter of amplicon size difference to determine a positive or negative, it is critical that this information is included. If it is a matter of amplification vs no amplification, then it should be states how erroneous amplification in non target species was mitigated by optimized PCR condition to obtain reliable results. This work may have already been performed by the authors, but as it does not appear in the manuscript it makes the data very difficult to interpret.

Otherwise, the manuscript is well written and addresses an important question. Should the authors sufficiently address my concerns, I would be happy to recommend this article for publication.

Author Response

This manuscript presented herein by Salina and colleagues represents an interesting investigation on whether centrifugation of milk samples to concentrate potential contaminating microbes prior to DNA extraction can improve the detection of Mastitis-associated Mollicute Species in cow milk. While this is certainly an important topic and question to be addressed, I unfortunately found the methods used to be a bit lacking, and insufficient to appropriately address this question (see comments below). As such, the results may have been skewed, not necessarily due to the absence of these species, but rater improper handling of the specimen that could result in false negatives. It is unclear if these issues are simply with insufficient reporting of methods used, or if no alternative methods were used altogether. Significant improvements need to be made to the methods before this manuscript can enter a publishable state. Below are my comments and recommendations:

Major points:

  • The authors aim to assess the efficiency of centrifugation in improving detection of Mollicutes in bovine milk, but do not offer a comparison between different centrifugation set ups, and instead use only a single centrifugation speed and time (5000xg for 30 minutes line 97 of manuscript).

Many issues here:

1a) How was the speed and time of centrifugation decided? If the authors performed a comparison to determine optimal time and speed of centrifugation those data should appear in this manuscript to justify why these conditions were chosen.

We thank the reviewer for the highlighted points and will answer questions 1) and 1a) below. The reference used for our study was added in line 97 of the manuscript.

We emphasize that in our study we used the methodology described by (PUNYAPORNWITHAYA et al., 2009) to increase the detection of Mycoplasma spp. in conventional PCR. Although the authors used the centrifugation technique with the aim of increasing the chance of growth of Mycoplasma spp. on Hayflick agar, we considered it appropriate to attempt to use centrifugation to extract DNA in our samples, which demonstrated to be favourable. Another point to be considered is that in our laboratory we have a centrifuge model for a volume of up to 50 mL of milk that reaches a maximum speed of 5000 g, so we considered using the same speed used in the study by (PUNYAPORNWITHAYA et al., 2009) in the previous centrifugation of 14 mL of milk.

1b) Mollicutes are particularly small, and not as dense as eubacteria given their lack of a cell wall. It is well understood within the Mycoplasmology community that speeds of roughly 10,000xg for at least 15 minutes at 4C are typically required to obtain sizeable pellets from centrifugation of many Mycoplasma species, and that is on liquid broth that is not as dense, complex, or as viscous as milk. It is likely, therefore, that if this was the only condition tested, that the centrifugation was insufficient to pellet and concentrate the mollicute species tested herein, potentially resulting in false negative results. If different centrifugation set ups were not assessed, this should be performed for the revision of this manuscript.

Thanks again for the reviewer's comment. We agree that the methodology presented in our manuscript lacked information regarding the centrifugation used for DNA extraction. Therefore, we have added this information to lines 99-108 for clarity for reviewers and readers.

The DNA extraction methodology used in the study presented here has already been previously published by our group (SALINA et al., 2020). In summary, the extraction consists of the initial centrifugation of 1mL of milk at 12000g for 10 min, followed by the pellet washing step, boiling at 100°C, and immersion in ice.

1c)The authors state in the methods (line 79), that milk samples were kept frozen at -20C prior to analysis. Mollicutes are particularly susceptible to loss of titer when frozen at temperatures below -80C (I have empirically tested this in the laboratory with multiple mycoplasma species and have found that when samples are frozen at -20C then thawed and assessed for viable counts you experience a reduction of roughly 8 logs of viable Mycoplasma counts vs a 1.5 to 2 log reduction when samples are frozen at -80C. If the mollicute cells were lysed due to storage conditions this could have further resulted in false negative detection results, especially since DNA definitely does not pellet with centrifugation at 5000xg for 30 minutes.

Thanks to the reviewer for the comment and suggestion based on his knowledge of mycoplasmas. We agree with the reviewer's comment about the limitation in relation to freezing and, for our next studies, mainly those that evaluate cell viability in growth in a culture medium, we will consider freezing all samples at -80°C.

As answered in item 1b) above, the preparation of the milk sample was carried out in two stages: the first with centrifugation at 5000g of a volume of around 14mL of milk, and resuspension of the pellet in 1 mL of buffered saline solution, followed by of centrifugation at 12000g. The second step was centrifugation at 12000g of 1 mL of milk. Therefore, we understand that the methodology and centrifugation at 12000g were adequate for the decantation of Mycoplasma spp. cells, as it was based on studies by other authors (PUNYAPORNWITHAYA et al., 2009).

2) The other major issue I have is with the method of detection itself (i.e. PCR set up used in the study). The authors do not describe multiple key details about the method, for example:

2a) Is the PCR with the primers provided supposed to result in amplicons with size differences to determine sample positivity/negativity, or is it a matter of amplification vs no amplification?  What does a positive vs a negative result look like? Gel images should be provided as data along with whether the sample was deemed positive/negative to support the manuscript.

We appreciate the reviewer's comment and suggestion. For the identification of positive samples, the presence of bands was observed in the agarose gel, stained with Nancy, and using a ladder of 100 base pairs as a molecular marker. This information is available on lines 129-132 of the manuscript. For further clarification on general PCR conditions, we have added a table on line 121.

2b) What regions in the mollicute genomes do the primers amplify, and what are the expected amplicon sizes? What are the optimal PCR conditions used in these reactions?

Thanks to the reviewer for the comment and suggestion. A table was added to the methodology with the requested information, containing sequences of the oligonucleotides, amplification cycle, and size of the products, and the references used (line 121).

2c) How are the primers validated? A quick NCBI blast of the universal mollicute primers finds 100% covarage on multiple other families of bacteria including Escherichia species, Achromobacter, Bordetella, Acinetobacter, Vibrio, and Bacillus species, and these with E values of 0.004 or less, and 100% sequence coverage. In silico PCR produces multiple viable amplicons from non Mollicute species when the listed primers are used, with variable sizes of amplicons. Is this the same for the species specific primers? Some detailed information about the PCR conditions used, and what defines a positive vs a negative is critical here. Should these primers not be specific, it is possible the positive results obtained may be false positives. The authors should expand the methods to include how these primers are validated. Again, if this is a matter of amplicon size difference to determine a positive or negative, it is critical that this information is included. If it is a matter of amplification vs no amplification, then it should be states how erroneous amplification in non target species was mitigated by optimized PCR condition to obtain reliable results. This work may have already been performed by the authors, but as it does not appear in the manuscript it makes the data very difficult to interpret.

We thank the reviewer for the comment and agree that information was missing for a better understanding of the methodology performed. To further clarify, we added the text on lines 115-119. The references, sequences, amplification cycles and size of the amplified products have been added in Table 1 of the manuscript (line 121).

Initially, PCR was performed with a primer that amplifies bacterias of the Mollicutes class. For each positive sample in the screening of the bulk tanks, a new PCR was performed individually for each of the species proposed in this study (M. bovis, M. bovigenitalium, M. alkalescens, M. bovirhinis, M. californicum). The decision to use the generic Mollicutes primer allowed us to screen the milk samples, reducing the number of analyses, since individual PCRs for each species were only performed on those positive for screening.

The sequences of the specific primers were submitted to BLAST to verify the amplified regions, ensuring that they were specific primers, according to the authors' work. (BOONYAYATRA et al., 2012; CHÁVEZ GONZÁLEZ et al., 1995; KOBAYASHI et al., 1998). According to (KOBAYASHI et al., 1998) the sequences used for the species of M. bovigenitalium and M. bovirhinis do not have amplification for other species, and this result was observed by us when using BLAST. The same can be concluded when we use the reference of (CHÁVEZ GONZÁLEZ et al., 1995) for the study of the species M. bovis, with specific amplification of this agent.

It should be noted that the fact that the generic primer for the Mollicutes class amplifies not only microorganisms of the genus Mycoplasma spp., being able to amplify Ureaplasma, Phytoplasma, Acholeplasma, was the factor that made us use these primers to carry out the screening, followed by the reactions in positive samples with species-specific oligonucleotides. Finally, we concluded that, if the reactions using specific primers showed a positive result, the occurrence of genetic material from the researched target species would be confirmed through genetic sequencing, as carried out in previous studies by our group (SALINA et al., 2020).

Otherwise, the manuscript is well written and addresses an important question. Should the authors sufficiently address my concerns, I would be happy to recommend this article for publication.

We thank the reviewer for his relevant suggestions and compliments and hope that we have answered your requests in the best way.

The authors.

References

BOONYAYATRA, S. et al. A PCR assay and PCR-restriction fragment length polymorphism combination identifying the 3 primary Mycoplasma species causing mastitis. Journal of Dairy Science, v. 95, n. 1, p. 196–205, 1 jan. 2012.

CHÁVEZ GONZÁLEZ, Y. R. et al. In vitro amplification of the 16S rRNA genes from Mycoplasma bovis and Mycoplasma agalactiae by PCR. Veterinary Microbiology, v. 47, n. 1–2, p. 183–190, 1995.

KOBAYASHI, H. et al. In Vitro Amplification of the 16S rRNA Genes from Mycoplasma bovirhinis, Mycoplasma alkalescens and Mycoplasma bovigenitalium by PCR. Journal of Veterinary Medical Science, v. 60, n. 12, p. 1299–1303, 1998.

PUNYAPORNWITHAYA, V. et al. The effect of centrifugation and resuspension on the recovery of Mycoplasma species from milk. Journal of Dairy Science, v. 92, n. 9, p. 4444–4447, 2009.

SALINA, A. et al. Microbiological and molecular detection of Mycoplasma bovis in milk samples from bovine clinical mastitis. Pesq. Vet. Bras, v. 40, n. 2, p. 82–87, 2020.

Round 2

Reviewer 1 Report

Authors have addressed the concerns

Authors have made the suggested changes

Author Response

Dear reviewer,

We appreciate your suggestions and the recommendation for the publication.

Best regards,
The authors.

Reviewer 2 Report

I thank the authors for taking the time to address my comments. While this is a massive improvement over the original manuscript, there remain a few other points that need to be addressed prior to publication. These are minor comments as they can be easily addressed, but are still absolutely critical and in my opinion MUST be addressed prior to publication.

My comments to the responses provided are as follows:

In response to point 1a): This is an acceptable response though unfortunately still does not detract from the fact the centrifugation steps may have not been sufficient to say with full confidence that the samples are negative, despite what the PCR results indicate. That is okay, but I recommend that a limitations section is added to the manuscript where this is discussed. Erroneous negative results are major motivation for you undergoing this work, especially as you state this in the first line of the discussion. Therefore it is important to acknowledge within the manuscript that this is still a limitation that exists in your work.

Furthermore, since you are using the work by PUNYAPORNWITHAYA et al., 2009 as a major basis for the manuscript, it should be explicitly stated in the methods that your centrifugation protocol was adapted from their work, rather than just providing a number citation after listing centrifugation. You can easily adapt and tweak the response you provided to me here to include on the manuscript.

While I do believe that the inclusion of this reference in the methods section in the original manuscript was likely a simple oversight, it does give the impression that it was done to elevate the novelty of your work, especially since the manuscript is written with the focus on centrifugation. It is important to give credit where credit is due, especially since their findings played a major role in determining your methods.

In response to point 1b): Thank you for adding this information, though my comment was not regarding the DNA extraction method, but simply the efficiency of the centrifugation in improving the detection of Mycoplasma species. This was partially addressed through your response to my first comment (1a). It is still great that this information was added, however, as it makes the methods more transparent.

In response to point 1c): No worries about storage conditions, you cannot go back and change how these samples were stored, but it should be considered for future work. As far as the centrifugation goes, see my comments to your response in 1a.

In response to all sections of point 2): I thank the authors for providing this information. However, this does not exactly address my comments still. After checking the species-specific primers and appropriate references it does appear that you are basing the results based on the presence/absence of amplification rather than amplicon size differences, at least for the species-specific PCR. This should be EXPLICITLY stated in the methods section.

As far as the universal primers go, in silico PCR does reveal that the primers can provide amplicons of variable sizes from non-mollicute species as well. Not just from Mycoplasma, Ureaplasma, Phytoplasma and Acholeplasma. They can amplify sequences from Escherichia, Achromobacter, Bordetella, Acinetobacter, Vibrio and Bacillus species as well, with amplicons of varying sizes. This means that unlike with the species-specific primers, your discrimination between a Mollicute positive vs negative PCR is not solely depended on whether there is an amplicon or not, but also one of a particular size. The reference you provided for these primers is quite old (1992), and it is possible they believed they were mollicute specific when the amount of available sequences was limited, but this may no longer be the case.

That is why it is very important to state explicitly in your methods how you decided what constituted a mollicute positive in the samples, given that DNA from unrelated bacteria can still produce a detectable amplicon. Representative gel images of positive and negative controls, and a few samples for each PCR result should be provided. With the currently provided information I cannot tell if your Mollicute positive samples are actually positive for Mollicutes, or other bacteria. Transparent methods, and gel images can go a long way in addressing these concerns.

Should the requested information be added to the manuscript, I would recommend it for publication

Author Response

Dear reviewer,

We provide the answers in an attached file.

Best regards,

The authors.

Round 3

Reviewer 2 Report

I want to thank the authors for addressing my concerns with the manuscript, and would like to recommend the article for publication.  Good work!